# Laparoscopic Vaginoplasty Procedure Using a Modified Peritoneal Pull-Down Technique with Uterine Strand Incision in Patients with Mayer–Rokitansky–Küster–Hauser Syndrome: Kisu Modification

**DOI:** 10.3390/jcm10235510

**Published:** 2021-11-25

**Authors:** Iori Kisu, Miho Iida, Kanako Nakamura, Kouji Banno, Tetsuro Shiraishi, Asahi Tokuoka, Keigo Yamaguchi, Kunio Tanaka, Moito Iijima, Hiroshi Senba, Kiyoko Matsuda, Nobumaru Hirao

**Affiliations:** 1Department of Obstetrics and Gynecology, Federation of National Public Service Personnel Mutual Aid Associations, Tachikawa Hospital, Tokyo 1908531, Japan; konakaramukana0101@hotmail.co.jp (K.N.); shiraishi15@hotmail.co.jp (T.S.); a.tokuoka97@gmail.com (A.T.); tpchkeigo@gmail.com (K.Y.); kunio.tanaka1017@gmail.com (K.T.); moito.not.for.sale@gmail.com (M.I.); semnamik@gmail.com (H.S.); umekiyo@hotmail.co.jp (K.M.); nobumaruhirao@gmail.com (N.H.); 2Department of Obstetrics and Gynecology, Keio University School of Medicine, Tokyo 1608582, Japan; mihoiida1029@gmail.com (M.I.); kbanno@keio.jp (K.B.)

**Keywords:** Davydov procedure, Mayer–Rokitansky–Küster–Hauser syndrome, neovagina, uterine factor infertility, uterus transplantation, vaginoplasty, vaginal agenesis

## Abstract

Various vaginoplasty procedures have been developed for patients with Mayer–Rokitansky–Küster–Hauser (MRKH) syndrome. Here, we describe a novel laparoscopic vaginoplasty procedure, known as the Kisu modification, using a pull-down technique of the peritoneal flaps with additional structural support to the neovaginal apex using the incised uterine strand in patients with MRKH syndrome. Ten patients with MRKH syndrome (mean age at surgery: 23.9 ± 6.5 years, mean postoperative follow-up period: 17.3 ± 3.7 months) underwent construction of a neovagina via laparoscopic vaginoplasty. All surgeries were performed successfully without complications. The mean neovaginal length at discharge was 10.3 ± 0.5 cm. Anatomical success was achieved in all patients, as two fingers were easily introduced, the neovagina was epithelialized, and the mean neovaginal length was 10.1 ± 1.0 cm 1 year postoperatively. No obliteration, granulation tissue formation at the neovaginal apex, or neovaginal prolapse was recorded. Five of the 10 patients attempted sexual intercourse and all five patients were satisfied with the sexual activity, indicating functional success. Although the number of cases in this case series is few, our favorable experience suggests that the Kisu modification of laparoscopic vaginoplasty procedure is an effective, feasible, and safe approach for neovaginal creation in patients with MRKH syndrome.

## 1. Introduction

Mayer–Rokitansky–Küster–Hauser (MRKH) syndrome is a congenital malformation characterized by the defective development of the Müllerian ducts resulting in the absence of a functional vagina and uterus in the presence of normally functioning ovaries. As a result, conception and vaginal sexual activity are compromised. In patients with MRKH syndrome, the creation of a neovagina allows for satisfactory sexual intercourse, and uterus transplantation (UTx) can allow patients with MRKH syndrome to give birth [1].

Several surgical and nonsurgical techniques for creating an adequately sized and functional neovagina to allow for sexual intercourse have been developed for patients with MRKH syndrome [2,3]. However, the surgical method with the best anatomical and functional outcomes is controversial due to a lack of comparative studies [2]. In addition, various materials have been used to cover the newly created space. The laparoscopic Davydov procedure is one of the most commonly used techniques, in which the vesio-rectal space is coated with peritoneum [4]. However, inadequate pull-down of the peritoneum, obliteration of the neovaginal vault or shortening of the neovagina, granulation tissue formation, and prolapse of the neovagina or bowel are potential limitations of this procedure [2,5,6]. Therefore, various modified laparoscopic Davydov procedures have been developed [6,7,8,9].

In this report, we describe a novel laparoscopic vaginoplasty procedure, known as the Kisu modification, using a pull-down technique of the peritoneal flaps with additional structural support to the neovaginal apex using the incised uterine strand (defined as the fibrous band that exists between and connects the bilateral uterine remnants) in patients with MRKH syndrome and report the anatomical and functional outcomes of our case series.

## 2. Materials and Methods

### 2.1. Patients

Ten patients with MRKH syndrome underwent neovaginal construction via laparoscopic vaginoplasty at our tertiary hospital. The study was approved by the Institutional Review Board of Federation of National Public Service Personnel Mutual Aid Associations, Tachikawa Hospital, and informed consent was obtained from all patients and/or their parents. All patients with primary amenorrhea underwent pelvic and abdominal ultrasonography, pelvic magnetic resonance imaging, hormonal profiling, and karyotyping to confirm the diagnosis of MRKH syndrome. The patients were counseled regarding the available management options, including surgical and nonsurgical techniques, and patients who chose to undergo a surgical procedure were enrolled in this study. Descriptive results are reported as means and standard deviations (mean ± SD).

### 2.2. Surgical Procedure

The laparoscopic vaginoplasty procedure used in this case series involved vaginal and laparoscopic approaches (Figure 1, Figure 2, Figure 3 and Appendix A).

After observing the pelvic and abdominal cavities via laparoscopy, an 8 cm needle was gradually inserted from the vaginal mucosa into the space between the bladder and rectum for injection of normal saline with adrenaline (1:200,000 dilution) to expand the rectovesical space. Then, a 1.5 cm incision was made transversely in the vaginal mucosa followed by gentle blunt dissection into the potential vaginal space between the bladder and rectum to the peritoneum under laparoscopic guidance to maintain the correct direction and avoid injury to the bladder and rectum. The peritoneum of the vesicouterine pouch and pouch of Douglas were mobilized as far as possible for ease of subsequent laparoscopic dissection to create the anterior and posterior flaps for the neovagina. 

Following the vaginal approach, a laparoscopic approach was used to dissect the supravesical peritoneum extensively along the uterine strand and the bilateral rudimentary uteri. The peritoneum was dissected from the bladder, allowing the anterior peritoneal flap to be pulled down to the vaginal introitus (Figure 1A and Figure 2A). Next, the uterine strand was lifted, and the peritoneum below the strand (in the pouch of Douglas) was incised transversely, dissected, and mobilized to create the posterior peritoneal flap (Figure 1A and Figure 2B). A 3 cm-wide mold was inserted through the dissected vaginal space and pushed into the vault, the bladder was dissected from the vault, and the apex of the vault was laparoscopically opened via a transverse incision below the uterine strand (Figure 1B and Figure 2C). A longitudinal incision was also made in the middle of the uterine strand, dividing it bilaterally (Figure 1C and Figure 2D). The edges of the separated anterior and posterior flaps were pulled down through the neovaginal canal under laparoscopic assistance and vaginally stitched to the anterior and posterior mucosa of the neovaginal introitus, respectively, using 3-0 absorbable interrupted sutures (Figure 1D and Figure 2E,F). The neovaginal apex was created by suturing the supravesical and suprarectal peritoneum using 1-0 absorbable interrupted sutures at the target neovaginal length (approximately 10 cm) while inserting the mold. The bilateral incised uterine strands were then sutured to the lateral aspects of the neovaginal apex using 1-0 absorbable interrupted or continuous sutures to cover the side wall of the neovaginal apex and fix the neovaginal canal in the pelvis to prevent prolapse of the neovagina (Figure 1F,G and Figure 3C,D). Finally, a methacrylic resin mold was inserted into the neovagina to prevent vaginal stenosis.

## 3. Results

The patient characteristics and postoperative outcomes shown in Table 1. These patients showed no associated anomalies, including those of the urinary, skeletal and recto-anal systems. One patient had a history of vaginal construction surgery at a different hospital. The mean patient age at surgery was 23.9 ± 6.5 years, and the mean postoperative follow-up period was 17.3 ± 3.7 months. The preoperative vaginal length ranged from absent to 2.5 cm, with the exception of the patient with a history of vaginoplasty who had a vaginal length of 4 cm. 

All patients had normally developed external genitalia, bilateral rudimentary uteri, a uterine strand, and normally developed ovaries and Fallopian tubes. A left functional rudimentary uterus was observed in two patients, which was excised. All surgeries were performed successfully without complications. The mean neovaginal length at discharge was 10.3 ± 0.5 cm. Anatomical success was achieved in all patients, as two fingers were easily introduced, the neovagina was epithelialized, and the mean neovaginal length was 10.1 ± 1.0 cm at 1 year postoperatively. No obliteration or granulation tissue formation of the neovaginal apex or neovaginal prolapse was recorded. (Figure 4).

Five patients had a sexual partner and attempted sexual intercourse. Although objective evaluation of a satisfactory sexual life was not performed in this study, these patients were satisfied with their sexual life, indicating functional success. Patients who were not sexually active continued intermittent dilator exercises that maintained an adequate length and neovaginal width (Appendix A).

## 4. Discussion

This report presents a novel laparoscopic vaginoplasty procedure using a pull-down technique of the peritoneal flaps with extra structural support to the neovaginal apex using the incised uterine strands in 10 patients with MRKH syndrome. 

Several nonsurgical and surgical procedures to create a neovagina for patients with MRHK syndrome have been developed. Primary vaginal self-dilation has been recommended as a first-line therapy as it is noninvasive, does not require hospitalization, has minimal complications, and is cost effective [2,3,5,10]. Although this nonsurgical procedure has a high success rate of 74–95% [2,3,11], it is time-consuming and requires strict patient compliance for daily dilation. In addition, dilation therapy may result in comparatively shorter vaginas than surgical procedures [2,3,12].

The surgical procedure that results in the best functional outcome and sexual satisfaction is controversial [2]. There are various surgical techniques for constructing a neovagina, and the choice is always a compromise between the individual needs of each patient and the surgeon’s expertise. The Davydov procedure is a classical and effective technique based on pulling down the peritoneum and suturing it to the vaginal introitus. This procedure is safe and easy to perform without the risk of peritoneal flap necrosis or rejection reactions, which are complications of other transferred grafts. Furthermore, epithelialization of the peritoneum begins immediately as the peritoneum has high regenerative powers and can undergo squamous metaplasia when exposed to the external environment, allowing for prompt sexual activity after surgery [6]. However, the Davydov procedure is limited by several potential complications [2,5,6]. Therefore, different methods for the mobilization and the pull-down technique of the pelvic peritoneum have been reported [6,7,8,9].

We previously reported a novel laparoscopic vaginoplasty procedure using a pull-down technique of the peritoneal flaps and the peritoneum of the supravesical pouch and pouch of Douglas with a transverse incision below the uterine strand and fixation of the incised uterine strand at the top of the neovagina [8]. This procedure allows the peritoneum to be pulled down to the vaginal introitus more easily and allows for the creation of a longer neovagina. This procedure also prevents stricture of the vaginal apex and prolapse of the neovagina, resulting in successful anatomical and functional outcomes. However, granulation tissue often forms at the neovaginal apex after surgery due to the existence of the strand at the top of the neovagina, and the neovaginal length depends on the position of the uterine strand even if the uterine strand can be placed at a higher level due to mobilization created by the transverse transection of the uterine strand. Therefore, improvements were necessary to achieve an optimal peritoneal vaginoplasty.

The Kisu modification described in this study improves our previously reported laparoscopic vaginoplasty procedure [8]. It consists of mobilization of the extensive anterior and posterior peritoneum in the supravesical pouch and pouch of Douglas, transverse and longitudinal incisions of the uterine strand, creation of the neovaginal apex at the target neovaginal length via suturing between the supravesical and suprarectal peritoneum, and bilateral fixation of the incised uterine strand with the neovaginal apex for additional structural support. This modification allows for the peritoneum to be pulled down to the vaginal introitus adequately and the creation of a neovagina with any length and width without granulation at the apex and prevents vaginal prolapse and perforation due to coitus by reinforcing the apex bilaterally with the incised uterine strand. 

The idea of extra structural support using incised uterine strands for the neovaginal vault in Kisu modification was inspired by the recipient surgery of UTx. The first UTx procedure was conducted in Saudi Arabia in 2000, but the transplanted uterus was removed after 99 days because of uterine prolapse with signs of necrosis and vascular thrombosis [13]. This made fixation of the transplanted uterus in the pelvis an important concern, and the method entailing the uterine strand being longitudinally cleaved and incised strands being laterally attached to the transplanted uterus for uterine fixation in the pelvis was developed for the current recipient surgery of UTx in MRKH syndrome [14]. We applied this technique for extra structural support to vaginoplasty to prevent neovaginal prolapse and reinforce the neovaginal apex. Another report also divided the rudimentary uteri to provide further support to the vaginal apex based on the same theory [7].

One potential limitation of the Kisu modification is that adhesions may form between the bladder and rectum as a result of suturing between the supravesical and suprarectal peritoneum. Adhesions may impair the feasibility of future UTx surgery due to altered pelvic anatomical structures. Vaginal dilation is an optimal option when subsequent UTx surgeries are planned [15]. Therefore, injuries to the bladder and rectum should be carefully monitored and avoided when patients plan to undergo UTx. UTx for women with uterine factor infertility is increasing; therefore, the demand for vaginoplasty for patients with MRKH syndrome is expected to increase in the future as the creation of an adequate neovagina is a prerequisite for UTx. As vaginal stenosis after UTx has been reported [16,17], surgical and non-surgical vaginoplasty procedures that can provide sufficient vaginal length and width are important for the reproductive plans of patients with MRKH syndrome. 

## 5. Conclusions

The reported outcomes of these 10 patients suggest that the Kisu modification of the laparoscopic vaginoplasty procedure is an anatomically and functionally effective surgical treatment for patients with MRKH syndrome. While this case series reports relatively few cases, the Kisu modification is a feasible peritoneal vaginoplasty technique that may achieve more optimal outcomes than current techniques for vaginoplasty.

## Figures and Tables

**Figure 1 jcm-10-05510-f001:**
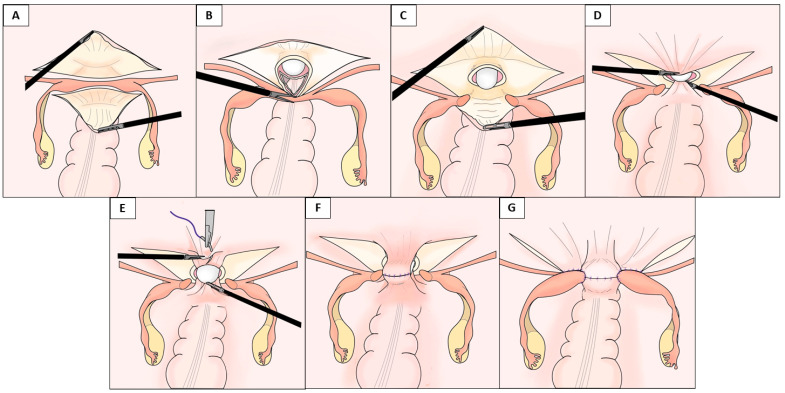
The Kisu modification. This schema shows the laparoscopic creation of a neovagina using a modified peritoneal pull-down technique with uterine strand incision. (**A**) The anterior and posterior peritoneal flaps (the peritoneum in the supravesical pouch and pouch of Douglas) are dissected extensively. (**B**) A transverse incision below the uterine strand serves as the opening of the neovaginal apex. (**C**) The uterine strand is divided via a longitudinal incision. (**D**) The anterior and posterior peritoneal flaps are pulled down through the neovaginal canal and sutured to the neovaginal introitus. (**E**) The neovaginal apex is created by suturing between the supravesical and suprarectal peritoneum at the target neovaginal length. (**F**) The neovaginal apex is shown before suturing the incised uterine strand to the lateral sides of the neovaginal apex. (**G**) The uterine strands provide additional structural support for the neovaginal vault.

**Figure 2 jcm-10-05510-f002:**
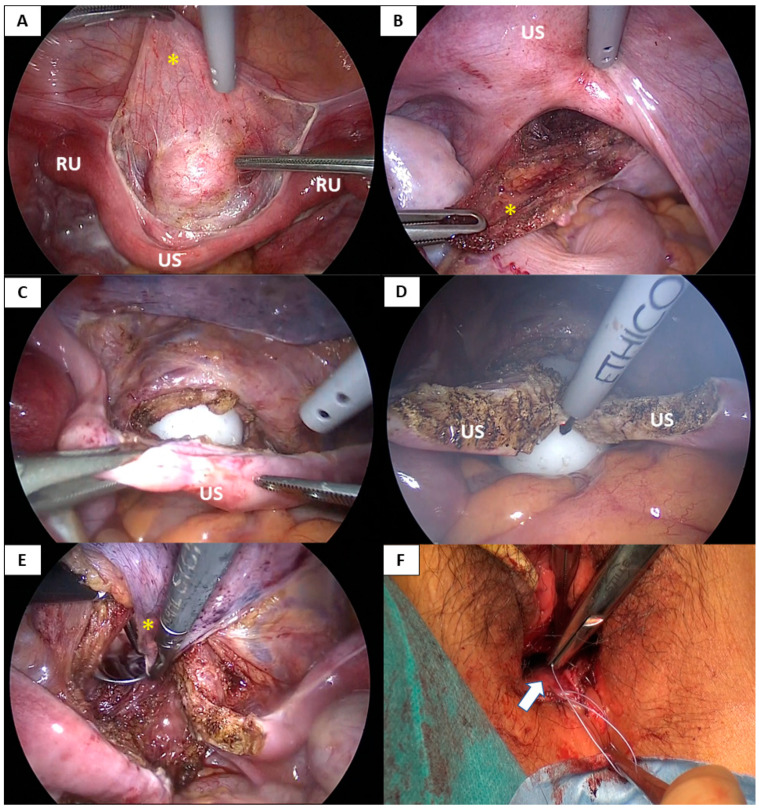
Laparoscopic pull-down technique of the peritoneal flaps with uterine strand incision. (**A**) The supravesical peritoneum (anterior peritoneal flap) (*) along the uterine strand and the bilateral rudimentary uteri is dissected from the bladder. (**B**) The peritoneum of the pouch of Douglas below the uterine strand is incised transversely and mobilized to create the posterior peritoneal flap (*). (**C**) A dilator is inserted through the dissected vaginal space and the apex of the vault is opened with a transverse incision below the strand. (**D**) A longitudinal incision is made in the middle of the uterine strand, dividing the uterine strand bilaterally. (**E**) The edges of the anterior (*) and posterior peritoneal flaps are pulled through the newly created canal under laparoscopic assistance. (**F**) The anterior and posterior flaps are sutured to the anterior and posterior mucosa of the neovaginal introitus. US; Uterine strand, RU; Rudimentary uterus.

**Figure 3 jcm-10-05510-f003:**
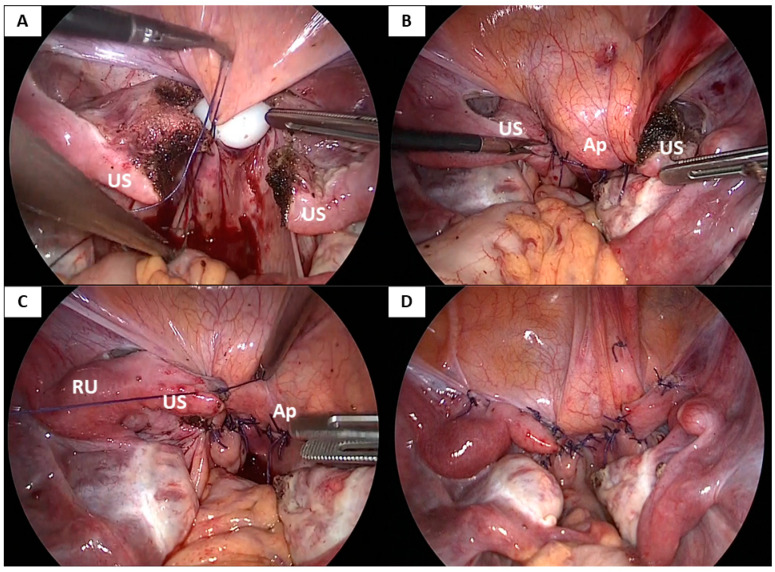
Creation of the neovaginal apex and additional structural support using the incised uterine strand. (**A**) The supravesical and suprarectal peritoneum are sutured at the target neovaginal length while inserting the mold. (**B**) The neovaginal apex is shown with the incised uterine strand before it is used to create additional structural support. (**C**) The incised uterine strand is sutured to the lateral side of the neovaginal apex to cover the side wall of the neovaginal apex and fix the neovaginal canal within the pelvis. (**D**) A final laparoscopic view of the neovaginal apex with the additional structural support created using the incised uterine strand in the Kisu modification technique. US; Uterine strand, RU; Rudimentary uterus, Ap: Apex of the neovagina.

**Figure 4 jcm-10-05510-f004:**
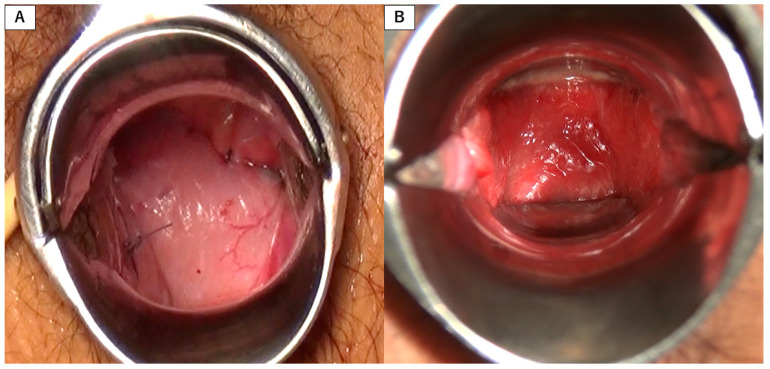
Speculum exam of the apex of the neovagina. (**A**) The neovaginal apex created by suturing the supravesical and suprarectal peritoneum is shown immediately postoperatively. (**B**) The mucosa of the neovagina is epithelialized without granulation tissue formation three months postoperatively.

**Table 1 jcm-10-05510-t001:** Postoperative outcomes of vaginoplasty in patients.

Case	Age at Surgery	Postoperative Follow-Up Length (Months)	Vaginal Length at Initial Examination (cm)	Neovaginal Length (cm)				Intraoperative Complications	Sexual Activity
				at Discharge	Three Months Postoperatively	6 Months Postoperatively	12 Months Postoperatively		
1	26	24	1	9	8	8	8	No	Satisfactory
2	21	23	Absent	10.5	10.5	10.5	9.5	No	Satisfactory
3	19	15	Absent	10.5	10.5	10	9	No	Satisfactory
4	23	16	Absent	10.5	10.5	10.5	10.5	No	Not attempted
5	34	14	4 *	10.5	11	11.0	10.5	No	Not attempted
6	18	15	Absent	10.5	10	9	9.5	No	Satisfactory
7	16	19	2.5	10.5	11	11	11	No	Not attempted
8	19	19	2	10.5	11	11	11	No	Not attempted
9	36	16	Absent	10.5	10.5	11	11	No	Not attempted
10	27	12	2	10.5	11	11	11	No	Satisfactory
Mean (±SD)	23.9 ± 6.5	17.3 ± 3.7	1.2 ± 1.3	10.3 ± 0.5	10.4 ± 0.9	10.3 ± 1.0	10.1 ± 1.0		

* History of vaginal surgery at an outside hospital.

## Data Availability

Data are available in a publicly accessible repository.

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
