# Peer review of "Laparoscopic Vaginoplasty Procedure Using a Modified Peritoneal Pull-Down Technique with Uterine Strand Incision in Patients with Mayer–Rokitansky–Küster–Hauser Syndrome: Kisu Modification"

_jcm, 2021, doi:10.3390/jcm10235510_

Round 1
Reviewer 1 Report
I read this paper with interest. It presents a surgical modification of interest to clinical practice. In the 10 cases presented, the length of the vagina was short before the operation, but it is unfortunate that the patients had the choice between a surgical technique and a non-surgical technique to obtain a satisfactory vaginal cup. The operated patients could have obtained the same results with a non-surgical technique.
Can the authors indicate the number of patients who had non-surgical management initially?
Author Response
Reviewer #1
Comment:
I read this paper with interest. It presents a surgical modification of interest to clinical practice. In the 10 cases presented, the length of the vagina was short before the operation, but it is unfortunate that the patients had the choice between a surgical technique and a non-surgical technique to obtain a satisfactory vaginal cup. The operated patients could have obtained the same results with a non-surgical technique.
Can the authors indicate the number of patients who had non-surgical management initially?
Response:
In this study, the patients who had non-surgical management before the operation were excluded. As the reviewer supports a non-surgical technique, primary vaginal self-dilation has been recommended as a first-line therapy as it is non-invasive, does not require hospitalization, has minimal complications, and is cost effective. Though this non-surgical procedure has a high success rate of 74-95% [2,3,12], it is time-consuming and requires strict patient compliance for daily dilation. In addition, dilation therapy may result in comparatively shorter vaginas than surgical procedures [2,3,13]. We have these descriptions in the Discussion of the original text, but we hope the reviewer understands them.
Reviewer 2 Report
MRKH syndrome is a very rare disease, so there have been few studies on its treatment. Davydov procedures are the most commonly performed treatments to improve the quality of life, and many studies on this will need to be conducted in the future. The authors' study of the novel laparoscopic vaginoplasty procedure has resolved these questions, and many researchers will have the opportunity to conduct further research.However, it seems that some corrections are needed.
1. In Table 1.
The 'Satisfactory of sexual activity' can be interpreted as having had coitus, but it can be interpreted as having a satisfactory sex life, so the terminology needs to be precise. If the author's intention is a satisfactory sex life, please add an additional explanation in the text with objective evaluation process.
2. Discussion, line 291
The authors stated that the new method has no complications compared to the Classic Davydov procedures. However, readers are unaware of what compliances with Classic Davydov procedures is. Above all, a detailed description of the complications with the classic surgical method is needed to explain the advantages of the new method.
3. Discussion, line 351
utors? Authors?.
4. Discussion, line 352
The findings and their implications should be discussed in the broadest context possible.
=> I understand the intention of the author, but please correct it with a clearer and more concise sentence.
Author Response
Reviewer #2
Comment:
MRKH syndrome is a very rare disease, so there have been few studies on its treatment. Davydov procedures are the most commonly performed treatments to improve the quality of life, and many studies on this will need to be conducted in the future. The authors' study of the novel laparoscopic vaginoplasty procedure has resolved these questions, and many researchers will have the opportunity to conduct further research.However, it seems that some corrections are needed.
Response:
We thank the reviewer for such a positive comment. We are also grateful to the reviewers for their critical comments and useful suggestions that have helped us in improving our manuscript considerably. As indicated in the responses that follow, we have taken all these comments and suggestions into consideration in the revised version of the manuscript.
Comment:
- In Table 1.
The 'Satisfactory of sexual activity' can be interpreted as having had coitus, but it can be interpreted as having a satisfactory sex life, so the terminology needs to be precise. If the author's intention is a satisfactory sex life, please add an additional explanation in the text with objective evaluation process.
Response:
We thank the reviewer for a constructive suggestion. We did not perform objective evaluations for sexual life, we have revised the description in Line 278 as follows. “Although objective evaluation of a satisfactory sexual life was not performed in this study, these patients were satisfied with their sexual life.”
Comment:
- Discussion, line 291
The authors stated that the new method has no complications compared to the Classic Davydov procedures. However, readers are unaware of what compliances with Classic Davydov procedures is. Above all, a detailed description of the complications with the classic surgical method is needed to explain the advantages of the new method.
Response:
We described the complications or disadvantages of the Classic Davydov procedures in the section of Introduction as follows; “However, inadequate pull-down of the peritoneum, obliteration of the neovaginal apex or shortening of the neovagina, granulation tissue formation, and prolapse of the neovagina or bowel are potential limitations of the Davydov procedure.” We hope these descriptions deserve an appropriate answer to this comment and the reviewer understands.
Comment:
- Discussion, line 351
utors? Authors?.
Response:
We apologize for the inappropriate descriptions in Line 351-354. These descriptions are parts of the instructions of the sample format and we forgot to delete these instructions before submission. We have deleted these sentences.
Comment:
- Discussion, line 352
The findings and their implications should be discussed in the broadest context possible.
=> I understand the intention of the author, but please correct it with a clearer and more concise sentence.
Response:
Same as above.